# Ion counting demonstrates a high electrostatic field generated by the nucleosome

**Magdalena Gebala[1]\*, Stephanie L Johnson[2], Geeta J Narlikar[2], Dan Herschlag[1,3,4]\***

[1]Department of Biochemistry, Stanford University, Stanford, United States; [2]Department of Biochemistry and Biophysics, University of California, San Francisco, San Francisco, United States; [3]Department of Chemistry, Stanford University, Stanford, United States; [4]ChEM-H Institute, Stanford University, Stanford, United States

**Abstract** In eukaryotes, a first step towards the nuclear DNA compaction process is the formation of a nucleosome, which is comprised of negatively charged DNA wrapped around a positively charged histone protein octamer. Often, it is assumed that the complexation of the DNA into the nucleosome completely attenuates the DNA charge and hence the electrostatic field generated by the molecule. In contrast, theoretical and computational studies suggest that the nucleosome retains a strong, negative electrostatic field. Despite their fundamental implications for chromatin organization and function, these opposing views of nucleosome electrostatics have not been experimentally tested. Herein, we directly measure nucleosome electrostatics and find that while nucleosome formation reduces the complex charge by half, the nucleosome nevertheless maintains a strong negative electrostatic field. Our studies highlight the importance of considering the polyelectrolyte nature of the nucleosome and its impact on processes ranging from factor binding to DNA compaction.

DOI: https://doi.org/10.7554/eLife.44993.001

**\*For correspondence:**
mgebala@stanford.edu (MG);
herschla@stanford.edu (DH)

## Introduction

The eukaryotic nuclear DNA forms a highly compact and organized structure referred to as chromatin. Despite its compaction, chromatin is accessible to a vast cohort of macromolecules which regulate its structure, dynamics, and structural plasticity and thereby influence gene expression and determine cell differentiation and state (*Hock et al., 2007*; *Luger et al., 2012*; *Keung et al., 2015*; *McGinty and Tan, 2015*).

The most basic level of nuclear DNA compaction is complexation with positively-charged histone proteins to form nucleosomes (*Figure 1A*). The nucleosome is composed of 147 base-paired (bp) DNA wrapped in a left-handed helix with ~1.7 superhelical turns around the core of eight histone proteins, two copies each of H2A, H2B, H3 and H4 (*Luger et al., 1997*; *Kornberg and Lorch, 1999*; *Richmond and Davey, 2003*). The DNA associates with the histone octamer via backbone and minor groove interactions that involve salt bridges, water-mediated and direct hydrogen bonds, and deep insertions of positively charged arginines into each DNA minor grooves facing the central histone octamer (*Richmond and Davey, 2003*; *Davey et al., 2002*; *Andrews and Luger, 2011*). DNA is one of the most charged polymers in nature, carrying two negative charges per base pair and it generates a strong negative electrostatic field (i.e., electrostatic force) that influences its mechanical properties and its interactions with proteins and small molecules (*Williams and Maher, 2000*). This field provides an enormous barrier to DNA compaction in the form of DNA/DNA self-repulsion

**Figure 1.** Nucleosome electrostatics: models and implications. (**A**) Schematic representation of nucleosome formation and DNA compaction. (**B**) Models of nucleosome electrostatics. According to Model I, the electrostatic field around the nucleosome is weak due to compensatory electrostatic interactions between the DNA and the positively charged histone octamer. In Model II the electrostatic field remains strong, as has been proposed by theoretical and computational studies. (*Schiessel, 2003*; *Materese et al., 2009*; *Elbahnsi et al., 2018*) (**C**) Schematic representation of the effect of nucleosome electrostatics on their propensity to compact according to Models I and II. The distance for full screening of electrostatic repulsion is considerably less for Model I (top) than for Model II (bottom; $r^I$ vs. $r^{II}$). (**D**) Schematic representation of an ion atmosphere around a low charge density (ρ) (left) and a high charge density molecule (right). (**E**) Fraction of associated counterions (e.g., cations around a negatively charged molecule) and coions (e.g., anions around a negatively charged molecule) within a molecule's ion atmosphere as a function of the molecule's charge density (assuming a uniform charge distribution for simplicity). The magnitude of the electrostatic field correlates with the charge density of the molecule: the higher the charge density, the larger magnitude of the electrostatic field and stronger counterion attraction, as depicted by the cartoons in part (**D**).
DOI: https://doi.org/10.7554/eLife.44993.002

(*Misra and Draper, 1999*; *Bai et al., 2005*; *Lipfert et al., 2014*). To compact the DNA the electrostatic repulsion needs to be mitigated through a process called electrostatic screening, which primarily occurs by attraction of positive charges like cations or positively charged proteins (*Lipfert et al., 2014*; *Anderson and Record, 1995*; *Draper et al., 2005*; *Draper, 2008*). Thus, the association of the DNA around the positively-charged histone octamer would appear to be an important step towards the electrostatic screening of negative charges on the DNA backbone. It is often implicitly assumed that the complexation of DNA into the nucleosome structure results in a complete electrostatic screening of the DNA and a weak electrostatic field surrounding the nucleosome, resulting in a lessened repulsion with other nucleosomes; we refer to this concept as Model I

in *Figure 1B*. While this assumption lacks a concrete theoretical basis, it appears to be broadly accepted in the research community, based on individual discussions and on feedback following presentations, although we have not found it explicitly stated in published work on DNA compaction and chromatin.

Current computational models (e.g., Poisson Boltzmann mean-field calculations and all-atom models) provide a diametrically opposing view of the nucleosome electrostatics (*Figure 1B*, Model II) (*Schiessel, 2003*; *Materese et al., 2009*; *Elbahnsi et al., 2018*). These models predict that geometric features such as the close wrapping of the DNA in the nucleosome results in enhanced local negative charge density (i.e., ρ, charge per volume) and additive electric field effects, referred as an electrostatic focusing, which in turn *increase* the electrostatic field, despite the lower net charge of the nucleosome relative to free DNA (*Schiessel, 2003*; *Materese et al., 2009*; *Elbahnsi et al., 2018*; *Rohs et al., 2009*; *West et al., 2010*). However, these models have yet to be experimentally tested.

Understanding nucleosome electrostatics is fundamental for understanding DNA compaction and interactions that regulate chromatin function and gene expression. Strong electrostatic repulsion would strongly oppose compaction (*Figure 1C*) (*Misra and Draper, 1999*; *Bai et al., 2005*; *Lipfert et al., 2014*). Yet, proteins that interact with DNA to control transcription, repair damage, remodel chromatin structure, and compact the chromatin by bridging nucleosomes often rely on electrostatic attraction and electrostatically-guided one-dimensional diffusion to locate binding sites and to bind DNA (*Shazman and Mandel-Gutfreund, 2008*; *Marcovitz and Levy, 2011*; *Hudson and Ortlund, 2014*). Notably, the energetics of these processes will strongly dependent of the magnitude of the electrostatic field generated by nucleosomes. Therefore, it is of fundamental importance to address the opposing concepts of nucleosome electrostatics existing in the research community (e.g. Model I and Model II) and to quantitatively dissect the nucleosome electrostatics by experimental approaches.

'Ion counting' is arguably the most effective experimental approach to analyze nucleic acid electrostatics and test theoretical predictions (*Bai et al., 2007*; *Pabit et al., 2010*; *Gebala et al., 2015*; *Gebala et al., 2016*; *Allred et al., 2017*; *Jacobson and Saleh, 2017*). It uses e̲quilibration with a b̲uffer solution followed by i̲nductively c̲oupled p̲lasma m̲ass s̲pectroscopy (BE-ICP MS) to precisely determine the number of ions that interact with a nucleic acid and form an ion atmosphere around the molecule—that is, the number of cations that are attracted to and anions that are repelled from the DNA over those present in bulk (*Figure 2*). These numbers are directly related to the magnitude of a molecule's electrostatic field and can thus be used to infer the strength of electrostatic interactions (see 'Strategy to measure the electrostatics of nucleosomes') (*Lipfert et al., 2014*; *Bai et al., 2007*; *Gebala et al., 2015*; *Gebala et al., 2016*; *Allred et al., 2017*; *Strauss et al., 1967*; *Manning, 1969a*; *Manning, 1978*; *Jayaram et al., 1989*; *Das et al., 2003*; *Muthukumar, 2004*; *Lyklema, 1995*; *Muthukumar, 2017*).

In this work, we use ion counting to determine the number of ions associated with free double-stranded (ds)DNA and with nucleosomes, providing a quantitative comparison of their net electrostatic fields. We find that canonical nucleosomes preferentially attract cations ('counterions') over anions ('coions') and do so to an extent similar to non-nucleosomal DNA, confirming the Model's II prediction of a strong negative field around nucleosomes. The studies presented herein are foundational for considering the physical and energetic basis for DNA compactions and chromatin organization as well as protein binding to nucleosomes and their subsequent functional consequences.

## Background
### Strategy to measure nucleosome electrostatics
Polyelectrolytes are polymeric macromolecules containing a large number of ionic or ionizable groups such as nucleic acids. Polyelectrolytes are surrounded by ions that fully counterbalance their charge, but Poisson Boltzmann (PB) electrostatic theory predicts that this charge balance is achieved differently for molecules of low vs. high charge density (i.e., the number of charges per given volume, or unit length), (*Figure 1D and E*) (*Anderson and Record, 1980*). Specifically, low charge density molecules achieve charge neutrality by equally attracting counterions (i.e., ions with charge opposite to the molecule) and excluding coions (i.e., ions with the same charge to the molecule) (*Figure 1D and E*, left). In contrast, molecules with high charge density and thus with strong

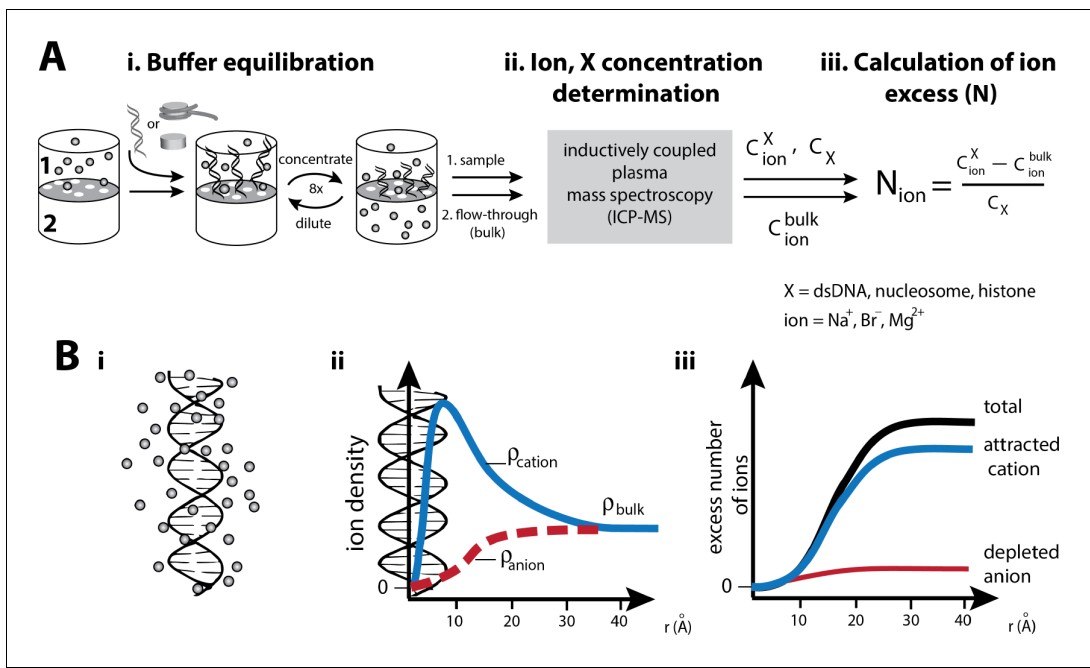

**Figure 2.** Experimental characterization of the ion atmosphere around macromolecules by 'ion counting'. (**A**) Schematic of an 'ion counting' experiment to quantify the composition of the ion atmosphere around dsDNA, nucleosomes, histones or the histone octamer (*Bai et al., 2007*; *Gebala et al., 2015*; *Greenfeld and Herschlag, 2009*). A detailed description of the ion counting method can be found in the Materials and methods section. Adapted from reference (*Gebala et al., 2015*). (**B**): i) Schematic representation of the cation excess surrounding a nucleic acid. ii) Schematic representation of ion density as a function of distance from a nucleic acid estimated by Poisson-Boltzmann (PB) theory; PB predictions adapted from (*Giambaşu et al., 2014*). Counterions and coions are radially distributed within the ion atmosphere around charged molecules. The highest density of counterions localizes near to the surface of the molecule, where the highest electrostatic potential resides; far from the molecule the density equals that of the bulk solution. Coions give the opposite behavior, with the lowest density near the molecule's surface due to electrostatic repulsion. iii) Computationally, the number of ions associated (the excess number, (**N**) with the ion atmosphere is calculated by integrating the excess ion density $(\rho_{DNA}^{ion} - \rho_{bulk}^{ion})$ around the nucleic acid (*Equation 4* in Material and methods).
DOI: https://doi.org/10.7554/eLife.44993.003

electrostatic fields, like DNA, are predicted to achieve charge neutrality by preferentially attracting counterions (cations for DNA) and excluding fewer coions (anions for DNA; *Figure 1D and E*, right); the strong electrostatic field of high charge density molecules can counteract the thermal motions of cations and result in their condensation around the molecules and hence a larger number of cations than anions around the molecule (*Muthukumar, 2004*; *Lyklema, 1995*; *Anderson and Record, 1980*; *Manning, 1977*; *Manning, 2002*; *Manning, 1969b*). This theoretically predicted preference is well-established by experiments (*Bai et al., 2007*; *Pabit et al., 2010*; *Gebala et al., 2015*; *Strauss et al., 1967*).

The degree of preference for counterion attraction varies continuously with charge density, as shown schematically in *Figure 1E*. Thus, the relative amount of counterion attraction and coion repulsion provides a measure of a molecule's net electrostatic field. For negatively charged molecules like DNA, we define β (*Equation 1*) as the fraction of charge neutralization that arises from association of cation (β+) vs. exclusion of anion (β–), where $N_{counterion}$ is the number of attracted counterions, $N_{coion}$ is the number of excluded coions, which can be experimentally determined by the ion counting methods (Figure 2A), and $q_{molecule}$ is the molecule charge. Because there is overall charge neutrality, the sum of the β values must be one (*Equation 1c*).

$$\beta_+ = \frac{N_{cation}}{q_{molecule}} \qquad (1a)$$

$$\beta_- = \frac{N_{anion}}{q_{molecule}} \tag{1b}$$

$$\beta_+ + \beta_- = 1 \tag{1c}$$

The salt concentration and the type of ion can also affect observed $\beta$ coefficients, but to lesser extent than the molecule's charge density. At high salt concentrations, one expects to observe a lesser attraction of counterions around and a stronger exclusion of coions from the charged molecules which is partly due to excluded volume effects (*Gebala et al., 2015*; *Anderson and Record, 1980*; *Giambaşu et al., 2014*), but still with asymmetric $\beta_+$ and $\beta_-$ values provided the polyelectrolyte is strong. Ions of higher valence are more effective in charge screening, hence their $\beta$ coefficients will be different than those for monovalent ions. The ion size will have insignificant effect on $\beta$s when only electrostatic interactions are the driving force for interactions, as we have previously shown for monovalent cations around dsDNA and dsRNA (*Gebala et al., 2016*). However, when ions form more specific interactions with molecules like inner-sphere or water-mediated coordination, the $\beta$ coefficients may differ for these ions.

The first method we use to test an overall electrostatic field around nucleosomes and their constituents (e.g. histones and DNA) is an explicit measurement of $\beta$ for cations and anions. Model I (*Figure 1B*), in the extreme, predicts a low electrostatic field resulting in equal cation attraction and anion repulsion around the nucleosome—that is, $\beta_+ = \beta_- \approx 0.5$ which is the lowest value for the counterion attraction. Model II predicts a strong electrostatic field, hence $\beta_+ > \beta_-$, with values similar to or more asymmetric than free dsDNA; (It is theoretically possible that $\beta_+ = 1$ yet, to our knowledge, this value (i.e., $\beta_+ = 1$) has never been reported; $\beta_+ = 1$ would suggest a process in which a charge molecule generates the electrostatic field capable of completely overcoming mixing entropy and driving a phase separation of counterions). To distinguish between Model I and Model II, we chose $\beta_+$ of dsDNA as a cutoff. The molecule (e.g., dsDNA) is considered to be one of the mostly charged polyelectrolytes generating one of the strongest electrostatic field amongst biological molecules. Previous studies have shown $\beta_+ = 0.81 \pm 0.02$ and $\beta_- = 0.19 \pm 0.03$ for short 24 bp DNA (*Bai et al., 2007*; *Gebala et al., 2015*).

A second way to test a molecule's net electrostatic field is to compare the attraction of counterions of different charge, such as $Mg^{2+}$ vs. $Na^+$ for dsDNA (*Gebala et al., 2016*; *Allred et al., 2017*; *Bai et al., 2008*; *Ni et al., 1999*). PB theory predicts that the preference for divalent over monovalent increases as the strength of the molecule's electrostatic field increases. For molecules generating weak electrostatic fields, the preference for $Mg^{2+}$ over $Na^+$ simply follows the bulk composition and ionic strength. For molecules generating strong electrostatic fields, the preference for $Mg^{2+}$ is greater, as each associated $Mg^{2+}$ can interact favorably with multiple closely spaced negative charges when a molecule has high charge density. This preference has also been experimentally verified for dsDNA (*Misra and Draper, 1999*; *Bai et al., 2007*; *Gebala et al., 2016*; *Misra and Draper, 2002*; *Misra and Draper, 2001*; *Xi et al., 2018*). Thus, the $Mg^{2+}$:$Na^+$ ratio provides a second measure of the electrostatic character of molecules (*Allred et al., 2017*; *Xi et al., 2018*).

## Ion counting method

The content of the ion atmosphere (e.g. the total excess number of ions (N) associated with the charged molecule with reference to the bulk solution) is experimentally determined by ion counting methods (*Bai et al., 2007*; *Pabit et al., 2010*; *Gebala et al., 2015*; *Gebala et al., 2016*; *Gross and Strauss, 1964*). Particularly effective is the ion counting through buffer-exchange inductively coupled plasma mass spectroscopy (BE-ICP MS); it allows the study of a large variety of ions over a broad range of concentrations, from tens of micromolar to molar, and can be carried out with high throughput to provide excellent precision and reliable statistics (*Gebala et al., 2016*; *Allred et al., 2017*). The method is comprised of three major steps: i) Buffer equilibration where the composition of the ion atmosphere of a given molecule (here nucleosomes, protein histones, and the dsDNA) is equilibrated against the bulk solution (*Figure 2Ai*). This step is carried out with centrifugal filters (Material and methods). ii) Analytical determination of ion concentration in a sample containing the molecule of interest (a sample in *Figure 2Ai*) and in the bulk solution (the 'flow-through' in *Figure 2Ai*) by inductively coupled plasma mass spectroscopy, ICP MS (*Figure 2Aii*). The method

allows the simultaneous determination of the nucleic acids concentration, by assaying the phosphorus content, although protein concentration is determined externally in the current experiments. iii) Calculation of the ion excess (N) around the molecule from ion concentrations measured by ICP MS (*Figure 2Aiii*).

BE-ICP MS has proven invaluable in testing electrostatic theories and computational models of the dsDNA and dsRNA specifically because it allows to make a direct comparison between experimental and computational data, obtained experimentally from the excess number of ions (N) in the DNA-containing sample relative to bulk (*Figure 2Biii*) and obtained computationally by integrating the excess ion density ($\rho_{ion}^{DNA} - \rho_{ion}^{bulk}$) around the nucleic acid (*Equation 4* in Material and methods) (*Bai et al., 2007*; *Gebala et al., 2015*; *Gebala et al., 2016*; *Allred et al., 2017*). Nevertheless, ion counting by BE-ICP MS has limitations. It delivers no information about the ion distribution within the ion atmosphere of macromolecules (*Figure 2Biii*) and hence no information about the ion atmosphere dimension or about variations of the electrostatic field around molecules— the shape and spatial extent of the ion atmosphere can be measure by anomalous small-angle X-ray scattering (ASAXS) technique. The method also counts ions within the ion atmosphere, yet at lower precision compared to BE-ICP MS (*Lipfert et al., 2014*; *Pabit et al., 2010*; *Das et al., 2003*; *Andresen et al., 2008*). Further, BE-ICP MS has limited capacity to accurately assay halogens, except $Br^-$ ions. For this reason, we chose to work with NaBr instead of NaCl. Our previous studies have shown that $Na^+$ accumulation around dsDNA is identical in the presence of $Cl^-$ or $Br^-$ over the concentration rage of 10-400 mM (and that $Cl^-$ or $Br^-$ exclusion are the same); in addition, NaCl and NaBr have similar activity coefficients (*Gebala et al., 2015*; *Gebala et al., 2016*). We also found no difference in RNA folding kinetics and thermodynamics in the presence of NaCl vs. NaBr (*Gebala et al., 2015*).

In summary, BE-ICP MS ion counting is a powerful and accurate tool to study molecular electrostatics; its characterization of global properties of the ion atmosphere provides information about the net electrostatic field surrounding a molecule, but not information about local variation of the field. Here, we partially compensate for this limitation by carrying out ion counting for full and truncated nucleosomes.

## Results

### Ion counting reveals that nucleosomes generate a strong negative electrostatic field

To determine the effect of nucleosome formation on DNA electrostatics, we experimentally measured the ions associated with free DNA and with nucleosomes by ion counting (*Figure 2A*) and from those values we calculated $\beta_+$ and $\beta_-$ for $Na^+$ and $Br^-$, respectively. Counting ions around 147 bp DNA revealed 5.4-fold preferential attraction of cations with respect to the anion exclusion, giving $\beta_+ = 0.85 \pm 0.02$, $\beta_- = 0.16 \pm 0.01$ (*Figure 3A—source data 1*). This large asymmetry in β coefficients is indicative of the strong negative electrostatic field generated by dsDNA.

We carried out the analogous experiment with nucleosomes reconstituted in vitro on the same 147 bp DNA (see Materials and methods). The total charge of the nucleosome (q), determined from ion counting (see Materials and methods, *Equation 3*), is considerably lower than that of the 147 bp DNA alone: $q_{Nuc} = -144.0 \pm 1.7$ vs. $q_{DNA} = -292.0 \pm 4.9$ for the nucleosome and the DNA, respectively. This decrease is expected from the association of the DNA with the positively charged histone octamer, and agrees quantitatively with estimates based on the histone amino acid compositions and PDB2PQR calculations (Total histone octamer charge: $+151 \pm 3.6$ (experimental, herein) and +149.0 e (calculated; see *Figure 3—source data 5* for calculations) (*Dolinsky et al., 2004*).

However, despite the overall reduction in charge by more than two-fold, the β coefficients for $Na^+$ attracted to and $Br^-$ excluded from nucleosomes remained similar to those for dsDNA alone; $\beta_+^{Nuc} = 0.83 \pm 0.020$ and $\beta_-^{Nuc} = = 0.17 \pm 0.015$ based on eight independent determinations (*Figure 3A and 3B*, *Figure 3—figure supplements 1* and *2*, and *Figure 3—source data 2*). These results provide strong evidence for Model II, which predicts that nucleosomes generate a strong negative electrostatic field.

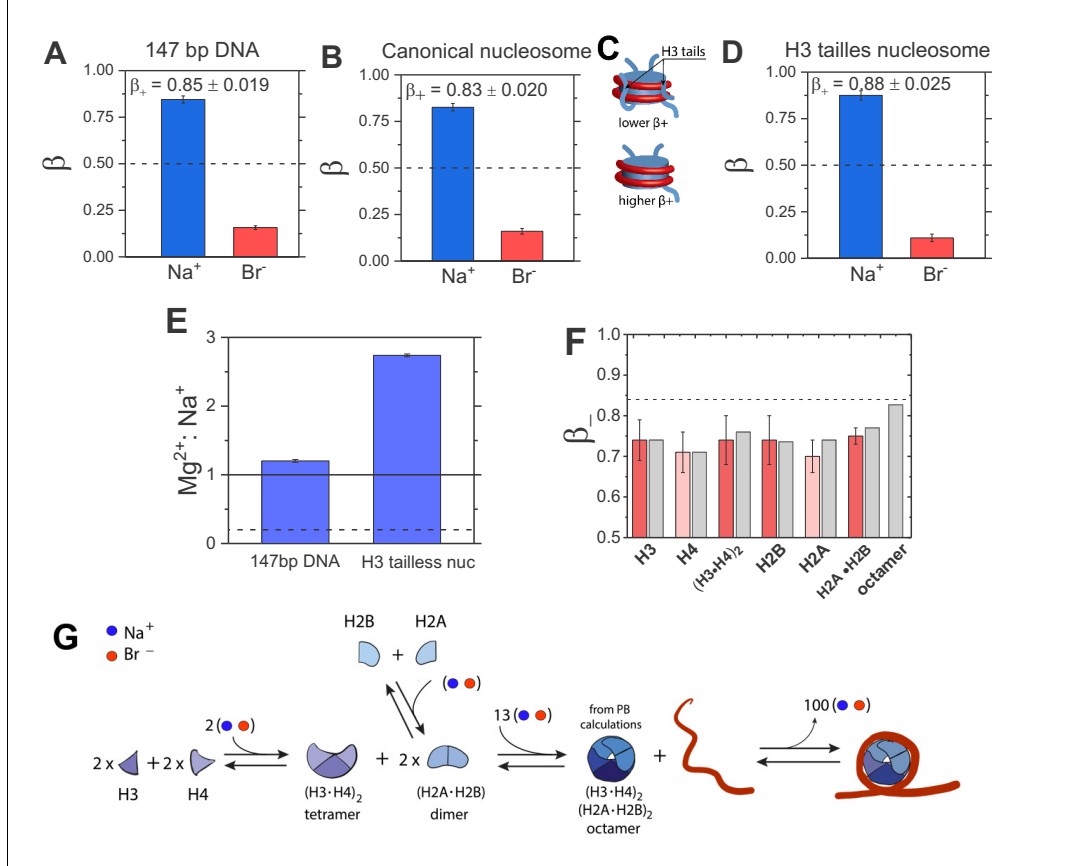

**Figure 3.** Quantification of electrostatic properties of a free dsDNA, nucleosomes, and histones by ion counting. (A) Experimentally-determined β coefficients of accumulated $Na^+$ cation (blue bar) and excluded $Br^-$ anions (red bar) around 147 bp DNA, (B) canonical nucleosomes, (D) H3 tailless nucleosomes. The $β_+$ values for the canonical and H3-tailless nucleosomes are significantly different based on two-sample t-test, with a p-value of 0.028. Dashed lines represent β = 0.5, the value predicted by the limiting case of Model I. The bulk concentration of NaBr was 10 mM in 2 mM Na-EPPS, pH 7.5 (E) $Mg^{2+}$ vs. $Na^+$ competition assay for 147 bp DNA and H3 tailless nucleosome. The solid line represents measurements for a model system, a short 24 bp DNA and the dashed line represents $Mg^{2+}:Na^+$ ratio for a low charge density molecule, an equivalent to β = 0.5. (F) $β_-$ coefficients for histone proteins and their complexes determined from ion counting results. Experimental results are shown in red bars and theoretical PB calculations are shown in gray bars. Ion counting experiments were carried out at 40 mM NaBr in 2 mM Na-EPPS, pH 7.5, whereas PB calculation were carried out at 40 mM monovalent salt. Dashed line represents the $β_+$ value for 147 bp DNA from *Figure 3A*. Each data point is the average of two repeats from at least two independent experiments. See *Figure 3A—source data 1-7* for raw data. (G) Schematic representation of octamer and nucleosome formation. Based on ion counting results and PB calculations for the histone octamer (*Figure 3A—source data 1-7*) we estimated the number of ions required at each step of the assembly, shown by the numbers in the assembly scheme. The formation of the octamer complex requires an uptake of ions to attenuate electrostatic repulsion between positively charged histone proteins, whereas ion release accompanies nucleosome formation. An equal number of cations and anions are taken up or released, as required (counterintuitively) to maintain charge neutrality (see *Lipfert et al., 2014*; *Misra and Draper, 2002*; *Draper, 2004* for explanation).

DOI: https://doi.org/10.7554/eLife.44993.004

The following source data, source code and figure supplements are available for figure 3:

**Source data 1.** Experimentally determined excess number ($N_i$), the $β_+$ coefficient (the faction of associated cations), and the $β_-$ coefficient (the faction of excluded anions)) for 10 mM NaBr around 147 bp DNA.
DOI: https://doi.org/10.7554/eLife.44993.008

**Source data 2.** Experimentally determined excess number ($N_i$), the $β_+$ coefficient (the faction of associated cations), and the $β_-$ coefficient (the faction of excluded anions) for 10 mM NaBr around canonical nucleosome.
DOI: https://doi.org/10.7554/eLife.44993.009

**Source data 3.** Experimentally determined excess number ($N_i$), the $β_+$ coefficient (the faction of associated cations), and the $β_-$ coefficient (the faction of excluded anions) for 10 mM NaBr around H3 tailless nucleosome.
DOI: https://doi.org/10.7554/eLife.44993.010

**Source data 4.** Experimentally determined excess number ($N_i$) for 25 mM NaBr and 2.5 mM $MgBr_2$ around 24 bp DNA, 147 bp DNA and H3 tailless nucleosome.

*Figure 3 continued on next page*

*Figure 3 continued*

DOI:

**Source data 5.** PB calculation of excess number ($N_i$) and the $\beta_-$ coefficient (the faction of associated anions), and the $\beta_+$ coefficient (the faction of excluded cation) for 40 mM NaBr around histone octamer.

DOI: https://doi.org/10.7554/eLife.44993.012

**Source data 6.** Experimentally determined excess number ($N_i$) and the $\beta_-$ coefficient (the faction of associated anions) and the $\beta_+$ coefficient (the faction of excluded cation) for 40 mM NaBr around histones H3, H4 and (H3·H4)$_2$ tetramer.

DOI: https://doi.org/10.7554/eLife.44993.013

**Source data 7.** Experimentally determined excess number ($N_i$) and the $\beta_-$ coefficient (the faction of associated anions) and the $\beta_+$ coefficient (the faction of excluded cation) for 40 mM NaBr around histones H2A, H2B and (H2A·H2B) dimer.

DOI: https://doi.org/10.7554/eLife.44993.014

**Source code 1.** Ion counting by Poisson Boltzmann.

DOI: https://doi.org/10.7554/eLife.44993.015

**Figure supplement 1.** Preferential association of ions (N) around a nucleosome and a function of the molecule concentration.

DOI: https://doi.org/10.7554/eLife.44993.005

**Figure supplement 2.** Non-denaturation PAGE (5%) showing nucleosomes before ion counting (1) and after ion counting experiments with 10 mM NaBr (2) and a mixture of 2.5 mM MgBr$_2$ and 25 mM NaBr (3) as well free 147 bp DNA (4).

DOI: https://doi.org/10.7554/eLife.44993.006

**Figure supplement 3.** Poisson-Boltzmann (PB) calculations of $\beta_+$ for Na$^+$ (**A**) and Mg$^{2+}$ and Na$^+$ competition (**B**) as a function of molecule charge density.

DOI: https://doi.org/10.7554/eLife.44993.007

## Removal of H3 histone tails increases the nucleosome electrostatic potential

The canonical nucleosome is comprised of histone proteins that have N- or C-terminal disordered and mobile extensions, referred to as tails, with a preponderance of positive charge. These tails form regions of positive electrostatic fields within nucleosomes and presumably can interact with the nucleosomal DNA and provide electrostatic screening of the DNA charge instead of cations (*Figure 3C*) (*Kan et al., 2007*; *Zheng and Hayes, 2003*; *Rhee et al., 2014*).

To test electrostatic effects of the tail, we reconstituted nucleosomes containing H3 histone proteins lacking their tails ('tailless'; see Materials and methods) and carried out the same analysis as we did for the canonical nucleosomes. The measured net charge of the H3-tailless nucleosome was more negative (e.g. q = –160 ± 3.1) and in good agreement with theoretical predictions; the estimated charge of the H3-tailless histone octamer is +132e (based on the amino acid composition PDB2PQR calculations), the charge of the 147 bp DNA is –292e, and thus the theoretical charge of the H3-tailless nucleosome equals –160e. Ion counting revealed larger $\beta_+$ and lower $\beta_-$ coefficients compared to values for the canonical nucleosome; $\beta_+ = 0.88 \pm 0.025$ and $\beta_- = 0.11 \pm 0.020$, respectively (*Figure 3B and D*, *Figure 3—source data 3*). The H3-tailless nucleosome attracts approximately 20% more Na$^+$ than the canonical nucleosome (e.g. $N_{Na}^{tailless} = 140 \pm 4.2$, $N_{Na}^{canonical} = 119 \pm 1.7$). Thus, this result shows that the net negative electrostatic field around nucleosomes increases when positively charged tails of the H3 histone are removed (*Figure 3A-D*). This result suggests that histone tails can make local contacts with the nucleosomal DNA (*Fletcher and Hansen, 1995*; *Hansen, 2002*; *Zheng et al., 2005*). Such contacts would mitigate some of the negative electrostatic field of the nucleosomal DNA, presumably in an asymmetric fashion corresponding to the positions of the tails within the nucleosomes.

## Mg$^{2+}$ vs. Na$^+$ competition substantiates a stronger electrostatic potential of nucleosome compared to free dsDNA

The increased preference for association with divalent over monovalent cations (M$^{2+}$ and M$^+$, respectively) as negative charge density increases provides a second measure of macromolecule electrostatics (see 'Strategy to measure nucleosome electrostatics' above). M$^{2+}$:M$^+$ competition is predicted by PB theory to depend linearly on a molecule's charge density ($\rho$). Hence, the cation competition is more sensitive to variations of molecule electrostatics than the $\beta$ values, which show exponential dependences on $\rho$ and thus have limited ability to resolve electrostatics of molecules generating strong electrostatic fields (see *Figure 3—figure supplement 3*).

Given that H3 tails partially counterbalance the negative field of the nucleosomal DNA, as observed above, we carried out this test with the H3-tailless nucleosomes. We previously measured equal amounts of $Mg^{2+}$ vs. $Na^+$ ($Mg^{2+}$: $Na^+$ ratio of $0.97 \pm 0.06$) around a 24 bp DNA despite a bulk concentration ratio of 1 $Mg^{2+}$ per 10 $Na^+$ (*Figure 3—source data 4*) (*Gebala et al., 2016*). A low charge density molecule that generates weak electrostatic field (Model I) would attract only 1 $Mg^{2+}$ for every 5 $Na^+$, under the same experimental conditions.

We found that the $Mg^{2+}$: $Na^+$ ratio around free 147 bp DNA was $1.15 \pm 0.02$ (95.0 $Mg^{2+}$ and 81.0 $Na^+$; *Figure 3—source data 4*), very similar to the value obtained for the 24 bp DNA (*Figure 3—source data 4*), under the same experimental conditions (2.5 mM $Mg^{2+}$ and 25 mM $Na^+$,-bulk concentrations). In contrast, for the H3 tailless nucleosomes, the ratio of associated $Mg^{2+}$: $Na^+$ was $2.75 \pm 0.17$, with $58 \pm 1.0$ $Mg^{2+}$ and $21 \pm 1.5$ $Na^+$ ions attracted to the molecule (*Figure 3E*). Thus, $Mg^{2+}$ vs. $Na^+$ competition provides additional support for Model II, that nucleosomes generate a strong negative electrostatic field. Furthermore, it suggests that the field increases upon complexation of dsDNA into nucleosomes, as more $Mg^{2+}$ cations are attracted to the nucleosome than to the free 147 bp DNA. Alternatively, some of the increased $Mg^{2+}$ attraction could arise from direct interactions with the nucleosomal DNA, which could reflect DNA structural rearrangements as well as increases electrostatic field (*Davey et al., 2002*; *Rohs et al., 2009*; *West et al., 2010*). The likely origin of the net high electrostatic field of the nucleosome is described in the Discussion.

## Histone proteins are positively charged but only partially attenuate the DNA electrostatic field

How does the overall electrostatic field of the nucleosome remain highly negative despite of the neutralization of half of the DNA overall charge by the positively charged histone octamer? To provide insights into this phenomenon, we determined the electrostatic fields generated by histone proteins and their stable sub-complexes (e.g. H2A·H2B dimer and $(H3·H4)_2$ tetramer), by quantifying their $\beta_+$ and $\beta_-$ values through ion counting. We measured on average 2.6-fold preferential attraction of anions with respect to cation exclusion for individual histone proteins ($\beta_+ = 0.25 \pm 0.05$–$0.31 \pm 0.1$ and $\beta_- = 0.69 \pm 0.06$–$0.74 \pm 0.05$, *Figure 3G*, *Figure 3—source datas 6* and *7*) and a small increase of anion attraction and decrease of cation repulsion for the H2A·H2B dimer and $(H3·H4)_2$ tetramer (e.g. on average $\beta_+ = 0.24 \pm 0.05$–$0.26 \pm 0.01$ and $\beta_- = 0.74 \pm 0.06$–$0.75 \pm 0.02$). These results indicate that histone proteins are positively charged, and they generate substantial electrostatic fields (i.e., $\beta_- > 0.5$), yet the fields are not as strong as dsDNA's ($\beta_- = 0.72 \pm 0.04$ on average for histones vs. $\beta_+ = 0.85 \pm 0.02$ for the dsDNA).

We also assessed the electrostatic field of the histone octamer core. However, as previous work has indicated that the octamer conformation is not stable under physiological or lower salt concentrations in the absence of DNA (*Rippe et al., 2007*) and because these conditions are required for ion counting experiments, we could only estimate the number of ions around the histone octamer through Poisson-Boltzmann (PB) calculations. To test this approach, we first compared the experimental and theoretical $\beta$- value of anion attraction for individual histone proteins and their stable sub-complexes and observed a good agreement for the predicted and measured values (*Figure 3F*).

Comparison of the PB calculations suggests that the octamer core attracts more anions than histone proteins alone (predicted $\beta_- = 0.84$ and $\beta_+ = 0.17$ vs. the predicted average histone $\beta_- = 0.73$) and thus that the histone octamer generates an electrostatic field comparable in strength but opposite in field to free dsDNA (i.e., the fraction of the octamer charge neutralization by anions ($\beta_- = 0.84$) is similar to the fraction of the DNA charge neutralization by cations ($\beta_+ = 0.85$). Indeed, our PB calculations suggest that the assembly of histones into octamers increases the electrostatic field around the octamer and that this process requires an uptake of approximately 13 ions to balance the increase (*Figure 3G*). Taken together, our findings raise the important question why histone octamers, despite their strong electrostatic fields, only partially attenuate the electrostatic field of DNA in nucleosomes, as evident from our ion counting measurements (*Figure 3B–3E*). We propose explanations for this phenomenon in the Discussion.

## Discussion

We have carried out the first experimental studies on the ion atmosphere around nucleosomes. We measured a similar degree of the cation attraction and the anion exclusion for the nucleosome as for

dsDNA. These results indicate that the net electrostatic field generated by the canonical nucleosome is similar to the field of DNA, despite the nearly two-fold decrease in the overall charge of the nucleosome relative to its free DNA. The observed strong cation association with nucleosomes and preferential association of $Mg^{2+}$ over $Na^+$ provide strong experimental support for prior computational and theoretic predictions (Model II, *Figure 1B*) (*Schiessel, 2003*; *Materese et al., 2009*; *Elbahnsi et al., 2018*) and raises important questions about both the nature of the strong overall electrostatic field of nucleosomes and how it affects DNA compaction and chromatin function. Our results counter the apparently popular perception that incorporation of DNA into a nucleosome leads to effective nullification of the DNA charge and electrostatic field.

## How does the net electrostatic field remain strong when the net charge of the nucleosome is much less than that of free DNA?

The simplest explanation for the maintained high electrostatic field comes from inspecting the nucleosome structure. Only about half of the DNA contacts the histone octamer, and the remaining half is exposed to solution and presumably subject to less extensive electrostatic screening from the octamer core (*Figure 4A*). Thus, one possibility is that the free part of the nucleosomal DNA behaves as a sheath with similar electrostatic properties as the free DNA and define the overall electrostatic character of the nucleosome. However, our ion counting studies on the H3-tailless nucleosome show that the electrostatic field of the nucleosomal DNA is even stronger compared to the free 147 bp DNA (*Figure 3D and E*), suggesting that this simple model is incomplete.

Electrostatic theory and computational studies provide a more comprehensive model (*Materese et al., 2009*; *Elbahnsi et al., 2018*). The magnitude of the electrostatic field generated by a charged molecule is determined by the molecule's charge density (i.e., the number of charges per given volume, surface area or unit length), rather than the net charge (*Muthukumar, 2004*; *Anderson and Record, 1980*; *Dill and Bromberg, 2012*). Thus, structural changes that increase the charge density also increase the electrostatic field, as observed when RNA molecules fold to compact three-dimensional structures (*Misra and Draper, 2002*; *Misra et al., 2003*; *Bonilla et al., 2017*). To qualitatively illustrate this model, we performed PB calculations of molecules surface potential, using the electrostatic energy at the distance of 1 Å from the molecule's surface (*Figure 4B*) (Notably, PB calculations have emerged as the approach of choice for assessing of the electrostatic properties of macromolecules, in part because PB is easily implementable, computationally tractable, and conceptually straightforward. However, PB calculations should be used with caution specifically when quantitative assessments on electrostatic energetics are made; the mean-field approximation and treatment of ions as point charges renders the PB model insufficient to fully account for the complexity of macromolecule electrostatic properties.) (*Giambaşu et al., 2014*; *Chen et al., 2009*) Wrapping the DNA around the histone core brings together backbone phosphoryl groups from distal parts of the DNA helix, and alterations in the duplex geometry decrease the distance between a subset of nearby phosphoryl groups (*Davey et al., 2002*). These features increase the negative electrostatic field, which is represented here by computed electrostatic surface potentials (*Figure 4Bii*). Association of the core only partially mitigates the increased electrostatic field, giving a final field that is more negative than the free linear dsDNA, in agreement with our experimental data (*Figure 4B,i vs. iii*). As nucleosomes are stable complexes, there must be a surplus of favorable DNA/nucleosome interactions to overcome, or 'pay for', the increased proximity of phosphoryl negative charges and resulting increased electrostatic field. Our ion counting experiments suggest that part of this energy arises from the significant release of ions accompanying the formation of the nucleosome (*Figure 3G*), consistent with the difficulty in obtaining equilibrium measurements for nucleosome formation and their extreme sensitivity to solution salt conditions (*Andrews and Luger, 2011*; *Thåström et al., 2004*; *Hazan et al., 2015*).

In principle, it is challenging to experimentally study electrostatics of heterogeneously charged molecules and determine local electrostatic fields, which play an important energetic role in processes that are dominated by short-range interactions like molecular recognition, folding or catalysis (*Honig and Nicholls, 1995*; *Davis and McCammon, 1990*). Experimental approaches that quantify amount of ions accumulated around charged molecules, like BE-ICP MS (*Bai et al., 2007*; *Gebala et al., 2015*; *Gebala et al., 2016*; *Das et al., 2005*) or SAXS (*Pabit et al., 2010*; *Das et al., 2003*; *Pabit et al., 2009*; *Qiu et al., 2007*) dissect the net electrostatic field—notably, a weak, net electrostatic field may result from compensatory effects of strong local electrostatic fields of similar

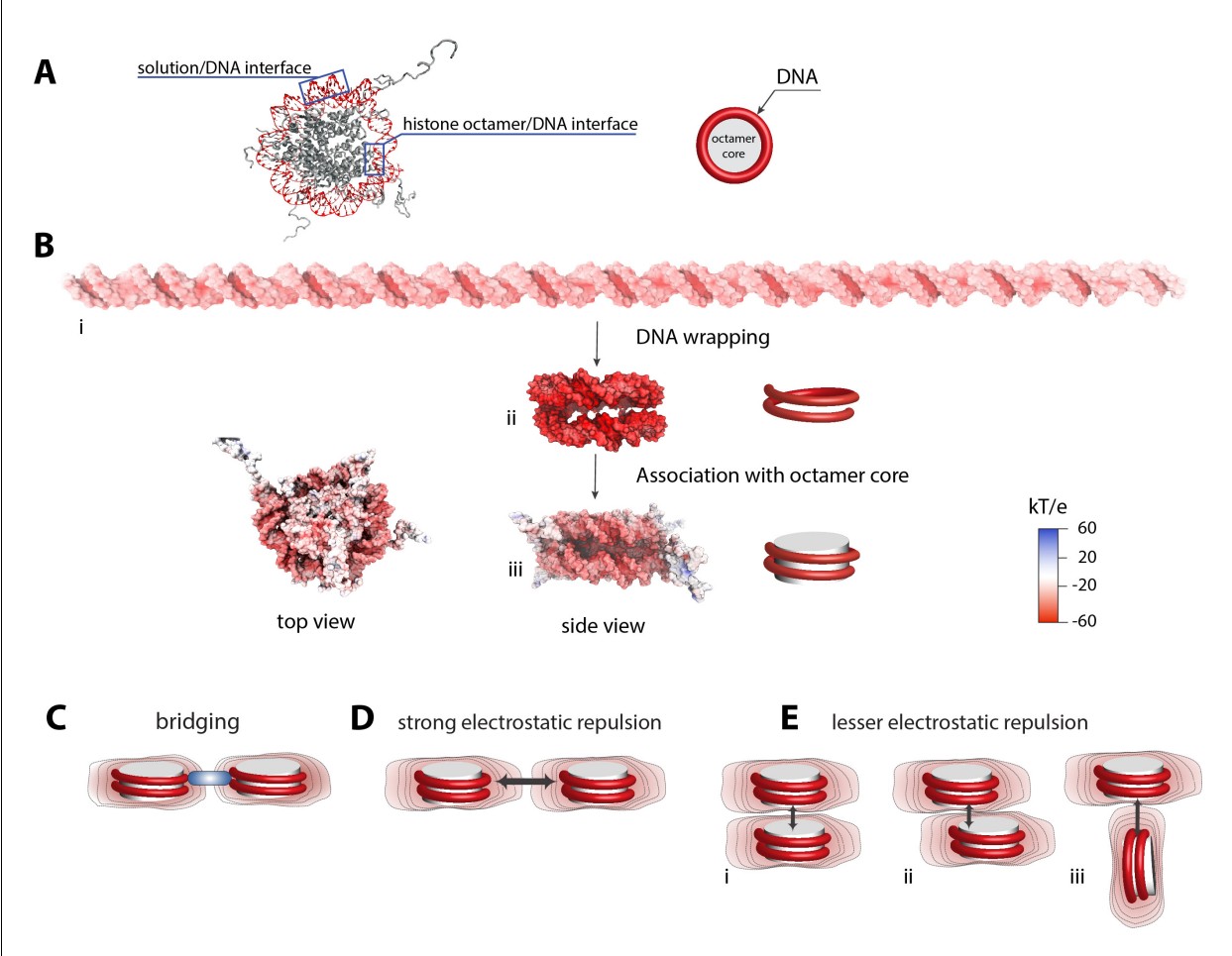

**Figure 4.** Electrostatic surface potential of a nucleosome and a dsDNA. (**A**) Crystal structure of a nucleosome (pdb 1kx5) (*Davey et al., 2002*). (**B**) Poisson-Boltzmann calculations of electrostatic surface potential of a dsDNA and a nucleosome (pdb 1kx5); the electrostatic potential ranges from +60 (blue) to −60 (red) $kTe^{-1}$, which highlights the difference in potential magnitude around the DNA and octamer core. The electrostatic potential mapped to the molecular surfaces was calculated using Adaptive Poisson-Boltzmann Solver (APBS) (*Dolinsky et al., 2004*) and the figures were rendered with VMD (*Humphrey et al., 1996*). The electrostatic surface potential of the nucleosomal DNA is largely negative and shown in red. (**C**)-(**E**) Schematic representation of electrostatic field lines around a nucleosome and models of internucleosomal interactions. The differences in the electrostatic field as shown in (**B**) may play an important role in internucleosomal interactions. High electrostatic potential of the nucleosomal DNA disfavors a side-by-side orientation as shown in (**D**) but introduce the possibility of bridging interactions stabilized by positively charged polyamines and proteins (C, shown in blue). Stacking of nucleosomes with the octamer-to-octamer orientation is energetically more favorable due to weaker electrostatic potentials at the octamer facets as shown in (**D**).

DOI: https://doi.org/10.7554/eLife.44993.016

The following figure supplement is available for figure 4:

**Figure supplement 1.** Poisson-Boltzmann calculations of electrostatic surface potential of a nucleosome particle (pdb 1kx5).

DOI: https://doi.org/10.7554/eLife.44993.017

magnitudes but inverse signs, whereas a strong net electrostatic field suggests even stronger local electrostatic fields. For the nucleosome, we measured a similar degree of cation association as we measured for the dsDNA that generates one of the strongest electrostatic field among biological molecules. Therefore, we concluded that nucleosomes also generate a strong net electrostatic field. Further, we hypothesized that local fields (e.g. specifically around the nucleosomal DNA) will be even stronger than the net or average value. To test this model, we carried out ion counting around H3-tailless nucleosomes—histone tails are positively charged and thus generate a local, positive electrostatic field. Indeed, the ion counting results revealed higher net electrostatic fields for tailless nucleosomes, supporting these predictions (we measured an increase in $\beta_+$ coefficient from 0.83 to

0.88). Importantly, next steps will be to measure a local electrostatic potential of the nucleosomal DNA, presumably using a similar approach proposed by Allred et al. to experimentally measure electrostatic potential of a short, 24 bp DNA (*Allred et al., 2017*)—the authors used the proton binding to $AH^+ \cdot C$ wobble pair, which is more favorable in the presence of the dsDNA compared to the same reaction occurring the bulk solution. Hence, energetical difference between these processes is used to determine the dsDNA electrostatic field.

## How does the high overall negative electrostatic field of nucleosomes affect DNA compaction and chromatin function?

It is frequently assumed that, because the positively charged histone octamer neutralizes some of the negative charge of DNA, histones must ameliorate repulsive interactions (referred here as Model I), thereby facilitating compaction into higher-order nucleosomal structures. However, when bringing charged macromolecules together, the repulsive interaction is not directly dependent on the net charge of the molecules but rather the strength of the electrostatic field at their surface. Our studies reveal that nucleosomes generate a net strong electrostatic field that may affect the interactions that are important for global and local organization of chromatin in the nucleus. Indeed, current computational studies have proposed that nucleosome electrostatics play an important energetic role in defining the orientation in which nucleosomes come together within an array (*Sun et al., 2005*; *Bascom et al., 2016*). As we show by PB calculation for individual nucleosomes, the strongest negative potential lies on the 'sides' (concentrated around dsDNA as it is generated by the molecule), so that side-by-side association would be less favored (*Figure 4D*). The negative potential is weakest at the 'top', and there is even a weak positive potential in the nucleosome center, where the histone core is not covered by DNA (*Figure 4B* iii and *Figure 4—figure supplement 1*, blue). These local differences in the nucleosome's electrostatic field suggest that there may be a strong tendency for nucleosomes to 'stack' off-center and to associate perpendicularly, as this would align regions of negative and positive potential (*Figure 4E*). Intriguingly, recent cryo-electron micrographs revealed preferential nucleosome association consistent with these electrostatic precepts (*Bilokapic et al., 2018*).

Discussions about the impact of nucleosome formation on DNA tend to focus on reduced DNA accessibility due to steric occlusion by the histone octamer (*Luger et al., 2012*; *McGinty and Tan, 2015*; *Zhu et al., 2018*; *Zhou et al., 2007*; *Struhl and Segal, 2013*; *Isaac et al., 2016*). However, the electrostatic properties of the nucleosome will also strongly influence how DNA interacts with proteins and small molecules and thus how chromatin compacts and functions. Importantly, our ion counting studies offer insights into the underlying mechanisms. Based on the observed stronger preferential association of $Mg^{2+}$ over $Na^+$ with the nucleosome compared to the free dsDNA, we propose that the higher electrostatic field and the positioning of phosphoryl oxygen atoms in nucleosomes may attract multivalent counterions (e.g. spermidine and spermine and DNA compacting proteins such as histone H1) more so than free dsDNA. These interactions in turn could promote bridging interactions between nucleosomes (*Figure 4C*) that lead to chromatin compaction. The positively charged histone tails have long been proposed to make bridging interactions between nucleosomes and are likely more effective than trans factors as they are pre-associated with the nucleosome (*Fletcher and Hansen, 1995*; *Hansen, 2002*; *Zheng et al., 2005*). Our studies suggest that such histone tail-mediated interactions also exploit the higher electrostatic field of a nucleosome.

Apart from such ordered interactions between nucleosomes, dynamic multivalent interactions have recently been implicated in heterochromatin formation by compacting nucleosome arrays into phase-separated, higher-order condensates (*Larson and Narlikar, 2018*; *Larson et al., 2017*; *Gibson et al., 2019*). Given the ability of nucleosomes to make strong electrostatic interactions with multivalent counterions demonstrated by our experiments and the long-range nature of these interactions, we hypothesize that nucleosome electrostatics also play a fundamental role in chromatin phase separation.

The non-uniform and concentrated electrostatic potential around the nucleosomal DNA likely not only plays an important role in organizing chromatin, but also in coordinating nucleosome-protein interactions that are at the heart of biological processes like gene transcription or DNA repair. Most DNA binding proteins are positively charged and their relative affinities are expected to be dependent on the local electrostatics of the nucleosome (*Rohs et al., 2009*; *Shazman and Mandel-*

*Gutfreund, 2008*; *Marcovitz and Levy, 2011*; *Hudson and Ortlund, 2014*; *Jones et al., 2003*; *Misra et al., 1994*). The strong and varied electrostatics of nucleosomes thus introduce an additional variable that nature has likely utilized in controlling gene expression. Dissecting this remains an important goal for future studies to fully and deeply understand the regulation and misregulation of gene expression.

# Materials and methods

**Key resources table**

| Reagent type (species) or resource | Designation | Source or reference | Identifiers | Additional information |
|---|---|---|---|---|
| Strain, strain background (*E. coli*) | BL21(DE3)pLysS | | Agilent #200132 | |
| Recombinant DNA reagent | Histone H2A expression plasmid | *Luger et al., 1997, Dyer et al., 2004* | | gift from Narlikar Lab |
| Recombinant DNA reagent | Histone H2B expression plasmid | *Luger et al., 1997, Dyer et al., 2004* | | gift from Narlikar Lab |
| Recombinant DNA reagent | Histone H4 expression plasmid | *Luger et al., 1997, Dyer et al., 2004* | | gift from Narlikar Lab |
| Recombinant DNA reagent | Histone H3 expression plasmid | *Luger et al., 1997, Dyer et al., 2004* | | gift from Narlikar Lab |
| Recombinant DNA reagent | Histone H3-tailless expression plasmid | *Luger et al., 1997, Dyer et al., 2004* | | gift from Narlikar Lab |
| Software, algorithm | apbs-pdb2pqr | http://www.poissonboltzmann.org/apbs/ | | |
| Software, algorithm | OriginPro 2017 | https://www.originlab.com | | |

## Reagents

DNA molecules to assembly 147 bp DNA (so-called 601 DNA sequence: CTGGAGAATCCCGGTC TGCAGGCCGCTCAATTGGTCGTAGACAGCTCTAGCACCGCTTAAACGCACGTACGCGCTGTCCC CCGCGTTTTAACCGCCAAGGGGATTACTCCCTAGTCTCCAGGCACGTGTCAGATATATACATCC TGT) were purchased from IDT (Ultramer DNA Oligonucleotide, Integrated DNA Technologies, USA). The purify of DNA (>96%) was verified by 5% native-PAGE gel with the load of 100–200 ng of DNA per lane, stained with SYBR Gold (Invitrogen, USA) with a DNA detection limit of 25 pg; CLIQS (Totallabs, UK) imaging analysis software was used for gel analysis. Histone expression plasmids were from Narlikar lab and BL21(DE3)pLysS competent *E. coli* cells were from Agilent Technology (USA). All salts were of the highest purity (TraceSELECT or BioXtra, Sigma-Aldrich USA). All solutions were prepared in high purity water, ultra-low TOC biological grade (Aqua Solutions, USA).

## Protein expression, purification and octamer assembly

All histones from *Xenopus leavis* (H2A, H2B, H4, H3 and tailless H3) were expressed from *E. coli* and purified following published protocols (*Guse et al., 2012*; *Dyer et al., 2004*; *Shahian and Narlikar, 2012*). The tailless H3 histone lacks the N-terminal region of the canonical H3 histone (25 residues including eight positively charged residues). Purification was carried out by an anion exchange through 5 ml HiTrap Q column followed by a cation exchange through 5 ml HiTrap S HP column. Subsequently, histones were subjected to gel filtration on a Superdex 75 column, to attain high purity. All columns were from GE Healthcare Life Sciences (USA). Histone octamer was assembled from purified histones as described (*Guse et al., 2012*; *Dyer et al., 2004*; *Shahian and Narlikar, 2012*).

## Nucleosome assembly

The 147 bp DNA was assembled from equimolar complementary strands (0.1–0.5 mM) in 100 mM Na-EPPS (sodium 4-(2-hydroxyehyl)piperazine-1-propanesulfonic acid), pH 7.5. Samples were incubated at 90℃ for 2 min and gradually cooled down to ambient temperature over 1 hr. Non-denaturing polyacrylamide gel electrophoresis showed no detectable single stranded DNA in samples, corresponding to >90% duplex; DNA stained by SybrGold (Invitrogen). Nucleosomes were assembled using published gradient dialysis-based protocols (*Guse et al., 2012*; *Dyer et al., 2004*; *Shahian and Narlikar, 2012*) and the purification of the nucleosomes was carried out on a 10% to 30% glycerol gradient. Subsequently, a fraction of collected nucleosomes was loaded onto 5% native-PAGE gel: the amount of nucleosome complex corresponded to >95% (DNA stained by SYBR Gold).

## Buffer Equilibration-Inductively coupled plasma mass spectroscopy (BE-ICP MS)

Buffer equilibration for nucleosomes and proteins was carried out following previous procedures (*Bai et al., 2007*; *Gebala et al., 2015*). NaBr and $MgBr_2$ samples were prepared in 2 mM Na-EPPS, pH 7.5 and their concentrations were determined by ICP MS. 500 uL–samples of nucleosome (4-12 µM) or proteins (4-100 ~ M) with the salt of interest were spun down to 100 µL at 7000 x g in Amicon Ultracel-30K filters (Millipore, MA) at 4℃ (*Figure 2*). Buffer equilibration was carried out until the ion concentration in the flow-through samples matched the ion concentration in the buffered solution used for the buffer exchanged (*Bai et al., 2007*; *Gebala et al., 2015*). No loss of the nucleosomes or proteins was observed during this procedure; no DNA or proteins were detected in flow-through samples, as determined by ICP MS, assaying the phosphorus content, or UV measuring absorbance at 280 nm. Nucleosomes were intact after the course of ion counting experiments as indicated by non-denaturation PAGE (*Figure 3—figure supplement 2*).

## Ion counting

Inductively coupled plasma mass spectrometry (ICP-MS) measurements were carried out using a XSERIES 2 ICP-MS (Thermo Scientific, USA). Herein, ion counting measurements were carried out with bromide salts, as the detection of $Br^-$ anion by ICP MS has highest accuracy and precision compared to other halogens (*Bu et al., 2003*). Aliquots (10–20 µL) of nucleosome- or histone-containing sample, the flow-through from the final equilibration, and the equilibration buffer were diluted to 5 mL in 15 mL Falcon tubes with water. Dilution factors, the ratio of diluted to total sample volume, were used to maintain sample concentrations within the linear dynamic range of detection (*Bai et al., 2007*; *Gebala et al., 2015*). Calibrations were carried out using standards from SpexCertiPrep (USA). Quality control samples, containing each element of interest at 100 µM, were assayed every ten samples to estimate measurement precision. To minimize memory effects in $Br^-$ detection, a solution of 5% ammonium hydroxide in highly pure, ion-free water (Mili Q) was used as a wash-out solution between measurements (*Bu et al., 2003*).

The count of associated ions around 147 bp DNA and nucleosomes is reported here as $N_i$ (e.g. the number of associated ions, i = + or –, indicating cation or anion, respectively) (*Anderson and Record, 1993*). The $N_i$ was calculated as the difference in the ion concentration between the equilibrated samples containing dsDNA ($C_{ion}^{dsDNA}$), nucleosome ($C_{ion}^{Nuc}$), or histones ($C_{ion}^{His}$) and the bulk solution ($C_i^{bulk}$), divided by the concentration of the molecules determined by phosphorous measurements using ICP MS (for the DNA and nucleosome) or determined by absorption at 280 nm for histones (*Equation 2*).

$$N_i = \frac{C_{ion}^{dsDNA} - C_{ion}^{bulk}}{C_{dsDNA}} \tag{2a}$$

$$N_i = \frac{C_{ion}^{Nuc} - C_{ion}^{bulk}}{C_{Nuc}} \tag{2b}$$

$$N_i = \frac{C_{ion}^{His} - C_{ion}^{bulk}}{C_{His}} \tag{2c}$$

For negatively charged molecules (e.g. DNA and nucleosomes), the $N_+$, is expected to be greater than zero, indicating their accumulation around the negatively charged polyelectrolytes, and $N_-$ for an anion is expected to be less than zero due to repulsive interactions with the DNA.

The total charge of the ionic species around molecules was calculated as the sum of the number of ions multiplied by their charge ($z_i$) and must counterbalance the molecule charge (q) (*Equation 3*).

$$total = \sum z_i N_i \tag{3a}$$

$$total = -q \tag{3b}$$

For each ion counting data point reported, at least two measurements were made on 2–4 different days with independently prepared samples. The reported errors are the standard deviations of all biological and technical replicates for a given sample. Data analysis was carried out using Origin-Pro2017 (OriginLab, Northampton, USA).

## Poisson Boltzmann (PB) calculations

NLPB calculations were carried out for a 24 bp DNA duplex, nucleosome and a sphere that was a test model. The B-form 147 bp DNA duplex was constructed with the Nucleic Acid Builder (NAB) package version 1.5 (*Bu et al., 2003*). PB calculation on nucleosome were carried out using X-ray crystal structure (pdb: 1kx5) (*Davey et al., 2002*). The 1kx5 file contains histone tails and we did not introduce any changes to the structure. Charges were assigned using the PDB2PQR routine with the Amber parameter (*Dolinsky et al., 2004*).

NLPB calculations were carried out using the Adaptive Poisson-Boltzmann Solver (APBS) (*Dolinsky et al., 2004*) on a 808 × 808×862 Å (*Keung et al., 2015*) grid with a grid spacing of 1.8 Å. The ion size equal 2 Å, the simulation temperature was set to 298.15 K and the dielectric constant of the solvent was set to 78. The internal dielectric was set to 2. The solvent-excluded volume of a molecule was defined with a solvent probe radius of 1.4 Å. Boundary conditions were obtained by Debye-Hückel approximation.

The number of ions of valence associated with macromolecules was computed by integrating the excess ion density: (*Shazman and Mandel-Gutfreund, 2008*; *Manning, 2002*; *Larson et al., 2017*; *Gibson et al., 2019*)

$$N_i = \rho_{b,i} \int \left( \lambda(r) e^{\frac{-z_i e\phi(r)}{kT}} - 1 \right) dr \tag{4}$$

where $\rho_{b,i}$ is the bulk ion density, $\lambda(r)$ is an accessibility factor that defines the region in space that are accessible to ions with $\lambda(r)$=1 and for the solvent-excluded region –that is, inside the macromolecule with $\lambda(r)$=0, $z_i$ is the elementary charge, $\phi(r)$ is the electrostatic potential, $k$ is the Boltzmann constant, and $T$ is the temperature.

The integration volume was defined as the entire volume of a simulation box including the solvent-excluded region in the molecule interior. Numerical integration of *Equation 4* was carried out using a custom written routine in C++, which is available for download (*Figure 3—source code 1*).

## Acknowledgements

The authors thank Bradley French and Broder Schmidt, members of the Herschlag and Straight lab for helpful discussions and critical advice. The authors thank Guangchao Li from Environmental Measurements Facility at Stanford University for outstanding technical assistance with ICP MS measurements.

## Additional information

### Competing interests

Geeta J Narlikar: Reviewing editor, *eLife*. The other authors declare that no competing interests exist.

## Funding

| Funder | Grant reference number | Author |
|---|---|---|
| National Institutes of Health | P01GM066275 | Dan Herschlag |

The funders had no role in study design, data collection and interpretation, or the decision to submit the work for publication.

## Author contributions

Magdalena Gebala, Conceptualization, Formal analysis, Visualization, Writing—original draft; Stephanie L Johnson, Geeta J Narlikar, Data curation, Methodology, Writing—review and editing; Dan Herschlag, Conceptualization, Data curation, Supervision, Funding acquisition, Writing—original draft

## Author ORCIDs

Magdalena Gebala (iD) https://orcid.org/0000-0002-1086-5548
Geeta J Narlikar (iD) http://orcid.org/0000-0002-1920-0147

## Decision letter and Author response

Decision letter https://doi.org/10.7554/eLife.44993.019
Author response https://doi.org/10.7554/eLife.44993.020

# Additional files

## Supplementary files

• Transparent reporting form
DOI: https://doi.org/10.7554/eLife.44993.018

## Data availability

All data generated or analyzed during this study are included in the manuscript and supporting files. Source data files have been provided for Figures 3 and 4.

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
