## [Decision Letter]

Thank you for submitting your article "Ion counting demonstrates a high electrostatic potential of the nucleosome" for consideration by *eLife*. Your article has been reviewed favorably by three peer reviewers, one of whom is a member of our Board of Reviewing Editors, and the evaluation has been overseen by John Kuriyan as the Senior Editor. The following individual involved in review of your submission has agreed to reveal their identity: Rohit V Pappu (Reviewer #3).

The reviewers have discussed the reviews with one another and the Reviewing Editor has drafted this decision to help you prepare a revised submission.

The reviewers and the editor find that your paper is a very valuable contribution to the field, but feel that the impact of the paper would be increased by addressing the points they have raised with textual revisions. Note that no new experimentation is called for, although a few new calculations may help increase the clarity of the conclusions.

Summary:

In this manuscript, Gebala et al. examine the electrostatic field of the nucleosome, the fundamental repeat unit of chromatin. The authors use a mass spectrometry-based approach to count and compare the number and nature (counter- versus co-ions) of ions associated with free DNA and with nucleosomes, which enables an elegant empirical quantification of the strength of the electrostatic field around the nucleosome. The authors discuss their measurements in light of two opposing models: model 1, where the wrapping of negatively charged DNA around the positively charged histone core would lessen the electrostatic repulsion of DNA in the context of an overall weakly charged nucleosome; and model 2, where the tight wrapping of DNA around the octamer would increase the electrostatic field (negative charge density) despite the reduced net negative charge of the nucleosome. The authors show that for the nucleosome, neutralization of negative charge arises predominantly from cation association (rather than anion exclusion) and that the nucleosome attracts preferentially bivalent Mg^2+^ rather than monovalent Na^+^ as counterions. Both observations argue for model 2, where the high negative electrostatic potential of DNA is retained even within the nucleosome context. Based on measurements of electrostatic potential with nucleosomes that lack the H3 histone tails, the authors suggest a role for histone tails in reducing the negative electrostatic potential of nucleosomal DNA and bridging interactions between nucleosomes.

This manuscript presents the first direct experimental approach to measuring electrostatics of the nucleosome in solution, which is an important contribution in itself given that electrostatics of the nucleosome greatly influence function, interactions, and structure of chromatin.

That being said, the following points noted below should be addressed in a revised manuscript.

Important points to address:

1) The distinction between models 1 and 2 seems to be artificial and the authors seem to have elevated a popular misconception about the nucleosome (that reviewers agree exists) into something approaching a concrete theory. They should not imply that other sources have advocated for this model unless they can provide a clear reference of that. While this notion is potentially corroborated by the quote from "Molecular Biology of the Cell" provided in the Introduction, it is not at all clear that this quote meant to suggest that the nucleosome completely neutralizes the charge of the DNA. More importantly, this does not accurately reflect current thinking in the field. In fact, there is a vast body of theoretical evidence from simulations and calculations that strongly support model 2, in agreement with electrostatic potential representations of the nucleosome structure. The authors accordingly cite several primary research articles supporting model 2, however only provide a single, textbook quote in favor of model 1. Moreover, the distinction between models 1 and 2 in terms of the measured parameters appears to be somewhat artificial (is there a cutoff β+ value above which one would favor model 2?). The authors need to be more circumspect in describing the current thinking in the field and models 1 and 2. The manuscript would benefit from a more nuanced narrative with more appropriate references and a focus on the measurements rather than perceived distinctions between models 1 and 2. It would appear that the data in this manuscript present an important contribution in themselves.

2) The paper could be much clearer in explaining and maintaining the distinctions between the overall charge, the charge density, the electrostatic field, and the electrostatic potential. More specifically, the descriptions of models for how histones might fail to weaken the electrostatic field of the DNA need to be updated, since at the moment they seem confusing and inaccurate.

The manuscript would benefit from a clarification of how the overall electrostatic potential relates to the local electrostatic potential (as calculated from PB, for example) as well as the net charge, and what predictions about e.g. protein binding can be made based on the knowledge of the overall electrostatic potential only. This would hopefully clarify why the authors state when introducing model 1, that abilities of DNA-binding proteins relying on electrostatics to bind DNA "would be lost if DNA's electrostatic field were nullified". This does not seem obvious in terms of overall electrostatic potential, which is used to distinguish between the two models. Do the authors imply here that a macromolecule/complex with a low overall electrostatic potential cannot efficiently engage in electrostatic interactions with other macromolecules? For example, could a protein that exhibits strong local electrostatic potentials of similar magnitudes but inverse signs on opposite sides (corresponding to a measured β+ = β- = 0.5) still bind DNA with its positively charged side?

It would be helpful if the authors provided, for the sake of comparison, a calculation of the expected β+ and β- if the charge was homogeneously distributed on the surface of the nucleosome.

3) The manuscript fails to convey to readers that β+ and β- depend on salt concentration and ion types. Suitable text should be added to insert caveats regarding the [salt] dependence of β+ vs. β-, thereby ensuring that the readers do not make erroneous generalizations. It would also be beneficial if the authors could add some data on the [salt] and ion type dependence of the measured values, but we appreciate the difficulties associated with performing such experiments.

4) The authors are arguing that the "overall electrostatic field" is the key parameter for assessing the extent that electrostatics will have an effect on DNA compaction as well as being the key parameter probed by ion counting, but fail to spell out what aspects of this 3D vector field they are referring to. More specific language is needed (e.g. the peak magnitude of the field near the DNA), along with some discussion of how ion-counting measurements would be affected around a complex with a heterogeneous field like the nucleosome.

To tie the argument that simple DNA compaction (i.e. without bridging) is not facilitated by the nucleosome, it would be helpful to provide some sort of numerical calculation to compare the strong repulsion shown in Figure 4D to the repulsion of bare DNA. Perhaps the authors can estimate the energetic penalty of bringing two histone-sized loops of DNA together compared to bringing the full nucleosomes together in this geometry.

5) Subsection “How does the high overall negative electrostatic potential of nucleosomes affect DNA compaction and chromatin function?”: "Our measurements and calculations also predict the topology type of contacts established in nucleosome arrays." The authors should more clearly state that their experimental measurements relate only to the overall electrostatic potential of the nucleosome and therefore cannot predict the preferred orientations of nucleosome-nucleosome interactions, whereas the local electrostatic potential distributions have been reported elsewhere previously. The authors should more thoroughly cite these papers (e.g., Sun, Zhang and Schlick, 2005) and discuss them in relation to nucleosome-nucleosome interactions.

6) The manuscript would benefit from a more in-depth description of the BE-ICP MS method as well as its benefits and limitations. Specifically, the authors should explain in more detail how the anions were measured, since usually they cannot be detected directly by this method. If possible, it would be helpful to test, whether the measured values would be different if NaCl was used instead of NaBr, or at least to discuss more extensively, whether the measurements could have been affected by the choice of NaBr.

7) It has been shown previously that PB calculations can give similar overall counterion accumulation / coion exclusion numbers as more elaborate MD simulations that explicitly account for ions and solvent molecules, but entirely different spatial distributions of ions (see http://dx.doi.org/10.1016/S0076-6879(09)69020-0). These calculations show that "trends in counterion accumulation as predicted by the PB calculations do not arise from the same regions as in the MD simulations". It is an important caveat that is worth noting inasmuch as the ion counting experiments are given a structural interpretation using PB calculations. The authors should insert appropriate cautionary language and citations with regards to the robustness of inferences drawn from PB calculations. This would highlight both the strengths and weaknesses of the ion counting experiments, which place constraints on our thinking but cannot go all the way and help us resolve between discrepant models for the structure-specific electrostatic fields / potentials that emerge from PB theory vs. other more explicit calculations. This shortcoming can be raised and summarized in the discussion in terms of issues that remain open.

---

## [Author Response]

This manuscript presents the first direct experimental approach to measuring electrostatics of the nucleosome in solution, which is an important contribution in itself given that electrostatics of the nucleosome greatly influence function, interactions, and structure of chromatin.That being said, the following points noted below should be addressed in a revised manuscript.Important points to address:1) The distinction between models 1 and 2 seems to be artificial and the authors seem to have elevated a popular misconception about the nucleosome (that reviewers agree exists) into something approaching a concrete theory. They should not imply that other sources have advocated for this model unless they can provide a clear reference of that. While this notion is potentially corroborated by the quote from "Molecular Biology of the Cell" provided in the Introduction, it is not at all clear that this quote meant to suggest that the nucleosome completely neutralizes the charge of the DNA. More importantly, this does not accurately reflect current thinking in the field. In fact, there is a vast body of theoretical evidence from simulations and calculations that strongly support model 2, in agreement with electrostatic potential representations of the nucleosome structure. The authors accordingly cite several primary research articles supporting model 2, however only provide a single, textbook quote in favor of model 1.

We rewrote the Introduction based on the reviewers’ suggestions. We now use language that does not suggest that the popular conception about the nucleosome electrostatics (referred in the submitted manuscript as Model I) has been proposed based on any concrete theory. Nevertheless, as the reviewers agree, this model is adopted implicitly in the research community, making it important to acknowledge and address in order to most clearly communicate our results to those working in this field.

Specifically, with respect to the Mode I we wrote: “It is often implicitly assumed that the complexation of DNA into the nucleosome structure results in a complete electrostatic screening of the DNA and a weak electrostatic field surrounding the nucleosome, resulting in a lessened repulsion with other nucleosomes; we refer to this concept as Model I in Figure 1B. While this assumption lacks a concrete theoretical basis, it appears to be broadly accepted in the research community, based on individual discussions and on feedback following presentations, although we have not found it explicitly stated in published work on DNA compaction and chromatin.”

With respect to the evidence for Model II, there had been no experimental tests, so while there may be a vast body of theoretical or computational results it is not clear whether one should consider this “evidence”. The remarkable lack of penetrance of these studies into the thinking of biologists presumably reflects some combination of the need for experimental tests and the foreignness of electrostatic theory to this group. We have worked hard in our presentation to respect the contributions from these disparate groups of scientists through our presentation and to synergize these fields through our experimental tests.

Moreover, the distinction between models 1 and 2 in terms of the measured parameters appears to be somewhat artificial (is there a cutoff β+ value above which one would favor model 2?). The authors need to be more circumspect in describing the current thinking in the field and models 1 and 2. The manuscript would benefit from a more nuanced narrative with more appropriate references and a focus on the measurements rather than perceived distinctions between models 1 and 2. It would appear that the data in this manuscript present an important contribution in themselves.

We agree that these differences are nuanced and that there is a continuous scale. Yet, it is helpful to the reader, and in particular the general reader, to lay out classes of models, while also noting the continuous nature of electrostatic fields, and we have done this in the revised manuscript, specifically in the section “Strategy to measure nucleosome electrostatics”.

Our treatment of this point in the section “Strategy to measure nucleosome electrostatics” considers ‘ends’ of the continuum as follows. The degree of the counterion attraction varies with the magnitude of the electrostatic field generated by charged molecules. In the case of the weak electrostatic field (Model I) β+ is approaching the value of 0.5, stated in the words β+ = 0.5 is the lowest value for the counterion attraction. On the other end of the scale would be β+= 1, which is theoretically possible yet, to our knowledge, this value (i.e., β+= 1) has never been reported; β+= 1 would suggest a process in which a charged molecule generates the electrostatic field capable of completely overcoming mixing entropy and driving a phase separation of counterions. In our work, to distinguish between Model I and Model II, we chose β+ for the dsDNA, as this molecule is considered to be one of the mostly charged polyelectrolyte generating one of the strongest electrostatic field amount biological molecules. The β+ for the dsDNA ranges from 0.81 ± 0.02 for short dsDNA (e.g. 24 bp) to 0.85 ± 0.02 for long dsDNA (e.g. 147 bp).

For the canonical nucleosome we measured β+ = 0.83 ± 0.02 and in the case of the H3-tailless nucleosome β+= 0.88 ± 0.025. Based on these values, we concluded that our results support Model II hypothesizing that nucleosomes generate a strong electrostatic field.

2) The paper could be much clearer in explaining and maintaining the distinctions between the overall charge, the charge density, the electrostatic field, and the electrostatic potential. More specifically, the descriptions of models for how histones might fail to weaken the electrostatic field of the DNA need to be updated, since at the moment they seem confusing and inaccurate.

We made the suggested changes and ensured that we used consistent terminology in the revised manuscript. Specifically, we emphasize that electrostatic field describes forces while electrostatic potential describes energy. We replaced the phrase overall charge with net charge as this better describes physical features of the nucleosome charge. We also made changes in the description of models for how histones might fail to weaken the electrostatic field of the DNA. We believe that these changes do indeed increase the manuscript’s clarity.

The manuscript would benefit from a clarification of how the overall electrostatic potential relates to the local electrostatic potential (as calculated from PB, for example) as well as the net charge, and what predictions about e.g. protein binding can be made based on the knowledge of the overall electrostatic potential only. This would hopefully clarify why the authors state when introducing model 1, that abilities of DNA-binding proteins relying on electrostatics to bind DNA "would be lost if DNA's electrostatic field were nullified". This does not seem obvious in terms of overall electrostatic potential, which is used to distinguish between the two models. Do the authors imply here that a macromolecule/complex with a low overall electrostatic potential cannot efficiently engage in electrostatic interactions with other macromolecules? For example, could a protein that exhibits strong local electrostatic potentials of similar magnitudes but inverse signs on opposite sides (corresponding to a measured β+ = β- = 0.5) still bind DNA with its positively charged side?

We agree that explanation of the difference between a global (overall) and local electrostatics is needed. First, we rewrote the Introduction and removed the statements that were misleading. Second, we added a paragraph in the Discussion explaining the inability of our approach to characterize electrostatic properties of charge-heterogenous molecules, in particular those which exhibit strong local electrostatic potentials of similar magnitudes but inverse signs in different places. We note that molecules exhibiting a weak net electrostatic field, may have strong local electrostatic fields and in this cases, electrostatic forces may play an important energic role in short-range interactions including binding and folding.

Nevertheless, we do, as we note, get some information about local electrostatic properties by comparing nucleosomes with and without the H3 positively-charged tails.

It would be helpful if the authors provided, for the sake of comparison, a calculation of the expected β+ and β- if the charge was homogeneously distributed on the surface of the nucleosome.

We agree, that models with uniform distributed charges are useful to build intuition. Specifically, models using well defined surface geometries are helpful to understand how β+ and β– values depend on the charge density of a molecule; β+ and β– are equal for low charge density and become asymmetrical at high charge density. Yet, the specific model with the uniform charges on the nucleosome’s surface would be less useful since the complex geometry of the nucleosome will have effects difficult to sort out and may not provide additional insights on nucleosome electrostatics for the general reader. We also worry that such a calculation, which has many assumptions and uncertainties, might be taken too seriously and might also detract from the simpler and more immediately compelling narrative of this manuscript. Instead, we have presented PB calculation on a sphere with homogeneously distributed charges as a model system and we compared how β+ and β– change with the charge density of the sphere (Figure 3—figure supplement 3). This figure shows that by adding negative charges on the surface of the sphere, without changing its geometry (e.g., a constant volume) the values of the β+ increased (e.g. approaches the value =1) as the charge density of the sphere increases.

3) The manuscript fails to convey to readers that β+ and β- depend on salt concentration and ion types. Suitable text should be added to insert caveats regarding the [salt] dependence of β+ vs. β-, thereby ensuring that the readers do not make erroneous generalizations.

We made the suggested changes in the section “Strategy to measure nucleosome electrostatics” on the dependence of βs on salt concentration, ion charge and type. In addition, we revised one of the supplementary figures (Figure 3—figure supplement 3) by including information on how bulk concentration affects β+ and β–. We stated in the revised manuscript that: “the salt concentration can also affect observed β coefficients, but to lesser extent than the molecule’s charge density. At high salt concentrations one expects to observe a lesser attraction of counterions around and a stronger exclusion of coions from the charged molecules which is partly due to excluded volume effects but still with asymmetric β+ and β– values provide the polyelectrolyte is strong.”

Importantly, the measurement of β+ and β– for nucleosomes at high salt concentration would not impact the general conclusions of this work.

It would also be beneficial if the authors could add some data on the [salt] and ion type dependence of the measured values, but we appreciate the difficulties associated with performing such experiments.

We showed the difference in association of monovalent (Na^+^) and divalent (Mg^2+^) cations around nucleosomes and this information illustrates that divalent cations interact more strongly than monovalent, as predicted for a strong polyelectrolyte. We have not carried out additional studies with different type of monovalent cations though we have shown in our previous work that association of the monovalent cations with the dsDNA and dsRNA is insensitive to variation of their size. While this may differ for nucleosomes, and be interesting, this is a distinct point from that of this manuscript and would require a large effort.

4) The authors are arguing that the "overall electrostatic field" is the key parameter for assessing the extent that electrostatics will have an effect on DNA compaction as well as being the key parameter probed by ion counting, but fail to spell out what aspects of this 3D vector field they are referring to. More specific language is needed (e.g. the peak magnitude of the field near the DNA), along with some discussion of how ion-counting measurements would be affected around a complex with a heterogeneous field like the nucleosome.To tie the argument that simple DNA compaction (i.e. without bridging) is not facilitated by the nucleosome, it would be helpful to provide some sort of numerical calculation to compare the strong repulsion shown in Figure 4D to the repulsion of bare DNA. Perhaps the authors can estimate the energetic penalty of bringing two histone-sized loops of DNA together compared to bringing the full nucleosomes together in this geometry.

We added text in Discussion explaining the limitation of ion counting to unequivocally address heterogenous electrostatic fields and local electrostatic fields. The emphasis of this work was on experimental determination of the net nucleosome electrostatics, as we could directly compare experimental and theoretical results. We measured a similar degree of cation association around the canonical nucleosome as we measured for the dsDNA, which generates one of the strongest electrostatic field among biological molecules. Hence, we concluded that nucleosomes also generate a strong global electrostatic field. Thus, if there is field heterogeneity, some local fields will be even stronger than the net or average value. Indeed, given what was known about histone “tails” we expected this to be the case and we could in part, test this possibility by removing H3 histone tails. Our ion counting results indeed revealed higher net electrostatic fields for tailless nucleosomes, supporting these predictions, and we have clarified these points in the revised text.

The reviewers suggested to estimate the energetic penalty of bringing two histone-sized loops of DNA together compared to bringing the full nucleosomes together in this geometry. We agree that it will be of great interest to model and experimentally address local electrostatics and their impact on compaction and complex formation, but we also view this as a distinct (important) research topic. Here, we have answered a fundamental question, providing first experimental test of common but largely overlooked electrostatic calculations by the research community. Notably, developing computational models on the nucleosome compaction likely requires better electrostatic models than PoissonBoltzmann, for reasons we discuss in the revised manuscript. We believe this should be the subject of distinct full study—one that will make new testable predictions.

5) Subsection “How does the high overall negative electrostatic potential of nucleosomes affect DNA compaction and chromatin function?”: "Our measurements and calculations also predict the topology type of contacts established in nucleosome arrays." The authors should more clearly state that their experimental measurements relate only to the overall electrostatic potential of the nucleosome and therefore cannot predict the preferred orientations of nucleosome-nucleosome interactions, whereas the local electrostatic potential distributions have been reported elsewhere previously. The authors should more thoroughly cite these papers (e.g., Sun, Zhang and Schlick, 2005) and discuss them in relation to nucleosome-nucleosome interactions.

We have modified this part in the Discussion based on reviewers’ comments and have added the suggested reference. We now emphasize throughout the text that ion counting measures the net electrostatic field of a molecule and has limited capacity to address local electrostatic fields.

6) The manuscript would benefit from a more in-depth description of the BE-ICP MS method as well as its benefits and limitations. Specifically, the authors should explain in more detail how the anions were measured, since usually they cannot be detected directly by this method. If possible, it would be helpful to test, whether the measured values would be different if NaCl was used instead of NaBr, or at least to discuss more extensively, whether the measurements could have been affected by the choice of NaBr.

We have added a section called “Ion counting” where we describe the method in detail. While there are now several papers on this approach, we agree that the general reader of this manuscript will not be familiar, and it is worth the space to provide this additional information in this manuscript. We explain that BE-ICP MS is limited in accurately determining halogens, except Br^–^. Therefore, we use NaBr instead of NaCl, and hence can explicitly measure the anion exclusion from the negatively charged molecules or anion association around the positively charged molecules.

Our previous, comprehensive ion counting studies on association of various monovalent salts around 24 bp DNA (Gebala et al., 2015) have shown that association of ions for NaCl and NaBr are the same within the experimental error. We also found no difference in RNA folding kinetics and thermodynamics in the presence of NaCl vs. NaBr.

7) It has been shown previously that PB calculations can give similar overall counterion accumulation / coion exclusion numbers as more elaborate MD simulations that explicitly account for ions and solvent molecules, but entirely different spatial distributions of ions (see http://dx.doi.org/10.1016/S0076-6879(09)69020-0). These calculations show that "trends in counterion accumulation as predicted by the PB calculations do not arise from the same regions as in the MD simulations". It is an important caveat that is worth noting inasmuch as the ion counting experiments are given a structural interpretation using PB calculations. The authors should insert appropriate cautionary language and citations with regards to the robustness of inferences drawn from PB calculations. This would highlight both the strengths and weaknesses of the ion counting experiments, which place constraints on our thinking but cannot go all the way and help us resolve between discrepant models for the structure-specific electrostatic fields / potentials that emerge from PB theory vs. other more explicit calculations. This shortcoming can be raised and summarized in the discussion in terms of issues that remain open.

We agree that the reviewers’ comment that good or excellent agreement between experimental and computational counts does not mean that any given computational model predicts well the energetics of molecules-ion interactions. There are, as the reviewers note, substantial differences between PB calculations and all-atom models like MD calculation on the ion distribution around charged molecules and hence energetics of the interactions. It is commonly assumed that all-atoms models more accurately account for energetics of nucleic-acid interactions and more accurately provide positional information compared to PB calculations, which are based on mean-field approximation and hence inherently incapable of fully accounting for the complex nature of the nucleic acid—ion interactions. Yet, the robustness of the all-atom calculations depends on multiple factors, specifically which type of forcefields are used. Our previous ion counting studies (Gebala et al., 2016) have shown that all-atom computational models do not accurately predict the energetics of ion association with dsDNA and dsRNA—i.e., their relative associations. Most generally, computational models should be tested against various experimental results and through providing blind predictions that can be orthogonally tested by experiments. We added discussion on this topic in the Discussion.